# Effect of Imipenem and Amikacin Combination against Multi-Drug Resistant *Pseudomonas aeruginosa*

**DOI:** 10.3390/antibiotics10111429

**Published:** 2021-11-22

**Authors:** Sara Mahmoud Farhan, Mohamed Raafat, Mohammed A. S. Abourehab, Rehab Mahmoud Abd El-Baky, Salah Abdalla, Ahmed Osama EL-Gendy, Ahmed Farag Azmy

**Affiliations:** 1Department of Microbiology and Immunology, Faculty of Pharmacy, Minia University, Minia 61519, Egypt; Sara.mahmoud@deraya.edu.eg; 2Department of Pharmacology and Toxicology, College of Pharmacy, Umm Al-Qura University, Makkah 21955, Saudi Arabia; maabdalla@uqu.edu.sa; 3Department of Pharmaceutics, College of Pharmacy, Umm Al-Qura University, Makkah 21955, Saudi Arabia; maabourehab@uqu.edu.sa; 4Department of Microbiology and Immunology, Faculty of Pharmacy, Deraya University, Minia 11566, Egypt; 5Department of Microbiology and Immunology, Faculty of Pharmacy, Suez Canal University, Ismalia 41522, Egypt; salah1979@hotmail.com; 6Department of Microbiology and Immunology, Faculty of Pharmacy, Beni-Suef University, Beni-Suef 62514, Egypt; Ahmed.elgendy@pharm.bsu.edu.eg (A.O.E.-G.); ahmed.abdelaziz@pharm.bsu.edu.eg (A.F.A.)

**Keywords:** *Pseudomonas aeruginosa*, drug combination, imipenem, amikacin, FICs, SEM

## Abstract

*Pseudomonas aeruginosa* is an opportunistic nosocomial pathogen associated with high morbidity and mortality rates. Combination of antibiotics has been found to combat multi-drug resistant or extensively drug resistance *P. aeruginosa*. In this study we investigate the in vitro and in vivo effect of amikacin and imipenem combination against resistant *P. aeruginosa*. The checkerboard technique and time-killing curve have been performed for in vitro studies showed synergistic effect for combination. A peritonitis mouse model has been used for evaluation of the therapeutic efficacy of this combination which confirmed this synergistic effect. The in vitro and in vivo techniques showed synergistic interaction between tested drugs with fractional inhibitory concentration indices (FICIs) of ≤0.5. Conventional PCR and quantitative real-time PCR techniques were used in molecular detection of *bla _IMP_* and *aac(6′)-Ib* as 35.5% and 42.2% of *P. aeruginosa* harbored *bla _IMP_* and *aac(6′)-Ib* respectively. Drug combination viewed statistically significant reduction in bacterial counts (*p* value < 0.5). The lowest *bla _IMP_* and *aac(6′)-Ib* expression was observed after treatment with 0.25 MIC of imipenem + 0.5 MIC of amikacin. Morphological changes in *P. aeruginosa* isolates were detected by scanning electron microscope (SEM) showing cell shrinkage and disruption in the outer membrane of *P. aeruginosa* that were more prominent with combination therapy than with monotherapy.

## 1. Introduction

*Pseudomonas aeruginosa* continues to be one of the most virulent opportunistic pathogens. It is recognized as a serious threat by the CDC [1]. This pathogen is considered as the leading cause of morbidity and mortality in immunocompromised hospitalized patients [2]. Nosocomial pneumonia, urinary tract infections, meningitis, endophthalmitis, external otitis, and endocarditis are the most common diseases caused by this pathogen. Immunocompromised neutropenic cancer patients, cystic fibrosis, bone marrow transplantation, and burn patients are the most susceptible to infection with *P. aeruginosa* [3].

Bacterial resistance to antibiotics has become one of the major worldwide problems in the past decade due to the increasing amount of extensively drug-resistant (XDR) and multidrug-resistant (MDR) pathogens and the decreased rate of development of new antibiotics [4]. Extensive drug resistance is found primarily in Gram-negative bacilli (GNB), especially *Enterobacteriaceae, P. aeruginosa, Acinetobacter baumannii*, and *Stenotrophomonas maltophilia* [5].

According to the Surveillance Network Database of *P. aeruginosa,* an increase in antibiotic resistance has been reported in recent decades, and it was considered a multi-drug-resistant pathogen. Thus, its eradication has become increasingly difficult due to its virulence mechanisms, its ability to resist antibiotics, its metabolic potential, and its physiological adaptability [6].

β-lactam antibiotics as carbapenems are considered as a drug of choice against susceptible *P. aeruginosa* and other Gram-negative bacteria due to their rapid bacterial killing [7]. However, in the last 10 to 15 years, carbapenem resistance disseminated worldwide, and the number of infections by these resistant isolates has increased [8]. Production of β-lactamases is considered the most common bacterial mechanism which counteracts β-lactams by hydrolyzing the β-lactam ring [9]. They are typically grouped into four distinct classes based upon DNA sequence. Molecular classes A–D. Molecular classes A, C, and D are serine-β-lactamases that include extended-spectrum β-lactamases (ESBL) that hydrolyze carbapenemases and cephalosporins such as *Klebsiella pneumoniae* carbapenemases (KPC), SHV, and CTX-M type. Class B called MBLs are Zn (II)-dependent enzymes that can accommodate most β-lactams in their active site and hydrolyze most β-lactam antibiotics as carbapenems, including: IMP and VIM groups, together with the emerging NDM group they commonly found in most *Enterobacteriacea* [10].

An aminoglycoside (AGS) as monotherapy can also cause bacterial killing against *P. aeruginosa*, but resistance has occurred against these drugs [11]. The most common mechanism of AG resistance is the chemical modification by aminoglycoside-modifying enzymes (AMEs). These enzymes are classified into three subclasses based on the type of chemical modification which apply to their AG substrates: AG N-acetyl-transferases (AACs), AG O-nucleotidyl-transferases (ANTs), and AG O-phospho-transferases (APHs). AACs that include four subgroups AAC(1), AAC(3), AAC(2′), and AAC(6′), selectively transfer an acetyl group from acetyl-CoA (Ac-CoA) to one of the several amine functions present in aminoglycosides [12].

Combination therapy is used in treating XDR and MDR infections, mostly in patients with immunodeficiency and/or repeated long-term use of broad-spectrum antimicrobial agents and in patients with severe underlying disease. This combined therapy could reduce the dose of antibiotics and decreases the emergence of resistance [5].

Synergistic combination therapy of antibiotics gives an attractive option to treat infections caused by multidrug-resistant *P. aeruginosa* [8]. β-lactam and aminoglycoside antibiotics have different mechanisms of action as there is no efflux pump which affects both antibiotics [13] and, aminoglycoside cause disruption in the outer membrane which enhances the target site penetration of β-lactams [14]

The purpose of this study is to evaluate the in vitro and in vivo efficacy of imipenem and amikacin combination against multi drug-resistant *P. aeruginosa* in comparison to the effect of each antibiotic alone.

## 2. Results

### 2.1. Isolation and Identification of Isolates

In the present study, out of 200 clinical samples collected from different clinical specimens, 150 isolates were Gram-negative rods found mostly in wounds 52%, chest infections 14.7%, ear infections 6.7%, burns 10%, urinary tract infections 8.7%, and gastroenteritis 8%. *E. coli* was the most common species found (60 isolates, 40%), followed by *P. aeruginosa* (45 isolates, 30%), 30 isolates of *Proteus* spp. (20%), 10 isolates of *Klebsiella* spp. (6.67%), and 5 isolates of *A. baumannii* (3.33%) Table 1.

### 2.2. Antibiotic Susceptibility Testing

Antimicrobial susceptibility testing showed that *P. aeruginosa* strains were highly resistant to ceftazidime (88.9%) and cefepime (82.2%) that was most prevalent among wound infections. Imipenem and amikacin were the most effective drug with least resistance percentage (28.9%) (Figure 1).

### 2.3. Detection of Imipenem and Amikacin MIC for Isolated P. aeruginosa Isolates

MIC values indicated that 28.9% of isolates were resistant to each amikacin (MIC ≥ 64) and imipenem (MIC ≥ 8). Amikacin MIC_90_ and MIC_50_ were 256 µg/mL and 8 µg/mL, while imipenem MIC_90_ and MIC_50_ were 256 µg/mL and 2 µg/mL (Table 2 and Table 3).

### 2.4. Molecular Detection of bla _IMP_ and aac(6′)-Ib by PCR

Sixteen isolates of *P. aeruginosa* harbored *bla _IMP_* (35.5%). On the other hand, 42.2% (19/45) of the isolates had *aac(6′)-Ib* gene. Four *P. aeruginosa* isolates (8.9%) harbored both genes. Appendix A indicate distribution of *bla _IMP_* and *aaac(6’)-Ib* genotype respectively among 45 isolated *P. aeruginosa*.

### 2.5. Determination of the Combined Effect between Amikacin and Imipenem against Resistant P. aeruginosa by Checkerboard Technique

The combined effect between amikacin and imipenem against selected strains of *P. aeruginosa* isolate wound no. 5 resistant for both drugs showed FIC_index_ ranging from 0.01 to 0.4, which meant that the combination had synergistic activity against the tested bacteria. MIC for amikacin and imipenem is greatly reduced from 512 µg/mL to 4 µg/mL and from 256 µg/mL to 1µg/mL, respectively with FIC_index_ 0.011 (Table 4).

### 2.6. Time–Kill Studies

Regarding *P. aeruginosa* isolate wound no. 5 resistant for both drugs, at 1 × MIC combination of each imipenem and amikacin, viable count was decreased by 38.9% (3.2 log_10_ CFU/mL) of the initial count after 24 h, which statistically differed from the reduction of each drug alone (*p* value ˂ 0.001). The observed reduction indicates bactericidal and synergistic effect of the two drugs (≥2 log_10_ CFU/mL reductions). The 2 × MIC and 4 × MIC combinations of each drug showed 30.3% reduction and total inhibition after 12 h, respectively (Appendix A and Figure 2A).

In *P. aeruginosa* isolate wound no. 2 resistant for imipenem only, reduction in CFU/mL was observed at the combination of 0.5 × MIC. The CFU/mL was reduced by 11.5% from the initial count after 24 h. At 1 × MIC combination the CFU/mL reduced by 50.2% from the initial count after 24 h, indicating synergistic and bactericidal effect of the combination. The CFU/mL was significantly reduced by the combination of both drugs than the imipenem alone (*p* value ˂ 0.001) (Appendix A and Figure 2B).

On the other hand, the *P. aeruginosa* isolate (wound no. 3) resistant for amikacin only showed a reduction in CFU/mL at 0.5 × MIC drug combination by 10.5% and at 1 × MIC combination by 48.8% from the initial counts after 24 h. The significant reduction in CFU/mL with drug combination indicated synergistic effect between both drugs (*p* value ˂ 0.001). Moreover, the combination showed a bactericidal effect after 24 h (Appendix A and Figure 2C).

### 2.7. Gene Expression 

Table 5 and Appendix A showed gene expression for selected resistance *P. aeruginosa* isolates using real-time PCR. The results showed over-expression of aac(6´)-Ib and *bla _IMP_* in the untreated *P. aeruginosa* isolates (wound no. 5), confirming the importance of resistance genes as one of the main resistance mechanisms in MDR or XDR *P. aeruginosa*. Moreover, the results showed that treatment with a combination of 0.25 × MIC imipenem + 0.5 × MIC of amikacin presented a decrease in fold change than 0.5 × MIC of imipenem + 0.25 × MIC of amikacin.

### 2.8. Scanning Electron Microscopy (SEM)

Effects of amikacin, imipenem, and combination of both drugs on the cellular structure of *P. aeruginosa* isolate wound no. 5 resistant for both drugs were confirmed by SEM analysis. The cells treated at 2 × MIC concentration of the tested antibiotics, showed altered morphology in the form of elongation and swollen cells in comparison to control. Non-treated bacteria were intact (regular rod-shaped) and showed a smooth surface, as can be seen in Figure 3A, while bacterial cells treated with the tested antibiotics showed considerable structural changes that caused pores on the outer membrane of *Pseudomonas* cells causing collapsing of the cells as can be obviously seen in Figure 3B,C. The combined antibiotics treatment altered the outer membrane, the structures of the cells and made them more permeable, Figure 3D.

### 2.9. In Vivo Studies

The in vivo results showed that the average blood bacterial counts were 10.04 log_10_ and 9.6 log_10_ CFU/ mL 3 h after infecting the mice (i.e., when the treatment started) for control and test groups, respectively. After 27 h, the positive control group showed elevation in bacterial count. On the other hand, the combination test group showed a significant reduction (*p* < 0.001) in the bacterial count by 39.6% from the initial count with high survival rates, which was more effective than imipenem and amikacin test groups. This indicated bactericidal and synergistic activity of the tested combination, Table 6 and Figure 4.

## 3. Discussion 

Treatment of *P. aeruginosa* infections has become one of the most difficult and problematic health issues worldwide because of the lack of effectiveness of some antipseudomonal antibiotics [2]. One of the common resistance mechanisms in *P. aeruginosa* was the acquisition of resistance genes by horizontal gene transfer leading to the emergence of MDR and XDR phenotypes which limited the therapeutic options of *P. aeruginosa* untreatable infections [2].

To the best of our knowledge, there are no data relating to whether a combination of imipenem with an aminoglycoside is better than imipenem alone against *P. aeruginosa*. So, this study is important due to the emergence of MDR, XDR, and PDR *P. aeruginosa* which is considered a great challenge to be treated by many antibiotics including imipenem. In addition, the pathogen nature, their intrinsic resistance to many antibiotics, and their ability to acquire multiple imported resistance mechanisms on mobile genetic elements or mutations lead to the increase of morbidity and mortality associated with their infections [15].

In the present study, out of 150 Gram-negative bacteria isolated during the study period, 30% of isolates was *P. aeruginosa.* A study done by [16] discussed that 3.8% of isolates were *P. aeruginosa;* another study done by [17] revealed that 6.4% of isolates was *Pseudomonas* spp.

*P. aeruginosa* isolates showed high resistance to ceftazidime 88.9% and cefepime 82.2%, while imipenem 28.9% was the most effective drug. A study done by [18] reported that *P. aeruginosa* isolates were resistant to all of the currently used antibiotics, including β- lactams (carbapenems and cephalosporins), aminoglycosides (gentamicin and amikacin), and fluoroquinolones (ciprofloxacin), yet remained susceptible to colistin.

The current study discussed that the no. of *P. aeruginosa*-harbored *bla _IMP_* was 16 isolates, 35.5% from total no. of *Pseudomonas*. Moreover, the current study showed that the number of *P. aeruginosa*-harbored *aac*(*6′*)*-Ib* was 19 isolates (42.2%). [19] reported that the most prevalent amynoglycosides-modifying enzyme was aac(6)’lb detected in a total of 72 isolates (36.0%), including 9 *P. aeruginosa*, *55 Enterobacterales*, and 8 *Acinetobacter spp.* In vivo model is a reliable tool for assessing the effectiveness of antibiotics on both susceptible and resistant strains of bacteria, besides confirming in vitro synergism and antagonism [20]. In our investigation, combinations of amikacin with imipenem reduce the susceptibility breakpoint in *P. aeruginosa* pathogen.

Our results revealed that the combination exhibited synergistic activity even at sub-inhibitory concentrations for imipenem-resistant or amikacin resistant *P. aeruginosa*. The study declared that the final bacterial counts decreased by combination better than each individual drug which was supported by in vivo experiment. Such combination may be a potential therapeutic option for the treatment of lethal infections caused by *P. aeruginosa* and can reduce the resistance risk of monotherapy with relieving the clinical treatment stress. [21] stated that gentamicin and meropenem combination was effective against all the *bla _IMP_*_-_producing *Enterobacteriaceae* and discussed strong bactericidal effects even at sub-MIC levels against strains resistant to both antimicrobial drugs by time killing curve.

The present in vivo study of mice after antibiotic therapy found a reduction in bacterial count that appeared in the group treated with imipenem and amikacin combination, which supports in vitro time killing curve. Uddin et al. [22] reported synergistic effect against *A. baumannii* by imipenem and amikacin combination. One of the mechanisms of *P*. *aeruginosa* resistance against imipenem and amikacin is the expression changes of *bla _IMP_* and *aac(6′)-Ib* genes responsible for the production of beta-lactamase and amino-glycosidase enzymes. Initially, *P. aeruginosa* resistant to imipenem and Amikacin were assessed via the conventional PCR technique. Furthermore, the expression changes of these genes (*bla _IMP_* and *aac(6′)-Ib*) were measured by qRT-PCR. The obtained results showed a decrease in gene expression by qRT-PCR after treatment with 0.25 MIC of imipenem + 0.5 MIC of amikacin combination than treatment with each antibiotic alone. We hypothesize that the changes in the expression of amikacin and imipenem resistant genes may be due to increasing the uptake of aminoglycosides by β-lactams. This finding was associated with an increased bactericidal rate and decreased cell replication [23].

SEM images showed that the combined imipenem and amikacin treatments altered the outer membrane integrity and made them more permeable due to the high affinity of imipenem toward penicillin-binding proteins (especially PBP2) [24]. In addition, binding of aminoglycosides to the negatively charged lipopolysaccharides in the outer membrane of *P. aeruginosa* disrupts the outer membrane structure [25]. This action occurs before they penetrate the cytosol and exert their intracellular effect on protein synthesis [26]. The permeabilizing effect enhances the periplasmic target site penetration of other antibiotics [27]. This result was agreed by [27,28].

## 4. Materials and Methods

### 4.1. Bacterial Isolates 

From February 2019 till December 2019, 200 clinical specimens were collected from patients admitted to El-Minia University hospitals. All types of infections were included in the study (wound, chest, ear, burns, UTI, and GIT infections). Methods of identification and confirmatory biochemical reactions are detailed in our previous published research [29]. The study protocol conformed to the ethical guidelines of the 1975, Declaration of Helsinki, as revealed in a priori approval (8/2021) by Ethical review board of faculty pharmacy, Deraya University, Egypt.

### 4.2. Antimicrobial Susceptibility Test

Antimicrobial susceptibility testing was performed by Kirby-Bauer disc diffusion method, Bauer, et al. [30], using 14 different antimicrobial agents according to the clinical and laboratory standard institute CLSI (2018) [31]. Antibiotics used were: Aztreonam (30 µg), Gentamycin (10 µg), Ceftazidime (30 µg), Cefepime(30 µg), Imipenem (10µg), Meropenem (10µg), Ciprofloxacin (5µg), Amikacin (30 µg), Piperacillin (100 µg), Norfloxacin (10 µg), Tobramycin (10 µg), and Levofloxacin (5 µg). In addition, all isolates were tested for minimum inhibitory concentrations (MICs) against both imipenem and amikacin by broth micro-dilution method according to the Clinical and Laboratory Standards Institute recommendations and interpretative criteria [31]. MIC for 90% of isolates (MIC_90_) and MIC for 50% of isolates (MIC_50_) was determined for better comparison.

### 4.3. Molecular Detection of bla _IMP_ and aac(6′)-Ib by PCR

Genomic DNA was extracted from overnight culture by a method described by Wilson [32]. The amplification was done using a 25 µL PCR mixture consisting of (200–400 ng) DNA sample, 12.5 µL PCR master mix 1 µL each of forward (20 pmol) and reverse primers (20 pmol), and nuclease-free water to 25 µL. PCR conditions were set to denaturation at 94 °C for 1 min; 35 cycles of denaturation at 94 °C for 30 s, annealing temperature according to Table 7 for 30 s, and extension at 72 °C for 1 min; and a final extension at 72 °C for 6 min a total of 35 cycles. Amplified product was separated on 2% agarose gel prepared in tris-borate EDTA buffer and stained with 10 μg/mL of ethidium bromide. DNA bands visualization using UV transilluminator were determined by visual inspection. Hundred-bp DNA ladder was used to assess the product size of *bla _IMP_* (488 bp) and product size of *aac(6′)-Ib* (365 bp).

### 4.4. Checkerboard Synergy Testing

Checkerboard synergism testing is considered the most standard technique used to determine the synergistic activity of antibiotic combinations. It is based on microdilution susceptibility testing of antibiotic combinations. The assays were performed with amikacin in combination with imipenem. Dilutions range from 64 to 0.03 µg/mL of each drug. The inoculum was prepared from colonies that had been grown on MHA overnight with OD 1.5 × 10^6^ cfu/mL. The in vitro interaction between these antibiotics was quantified by fractional inhibitory concentration (FIC)**.** The FIC index (FICI) was calculated using the following formula:FIC= FIC of drug A + FIC of drug B
FIC of drug A = MIC of drug A in combination/MIC of drug A alone.
FIC of drug B = MIC of drug B in combination/MIC of drug B alone.

Synergism is shown as FIC index of ≤0.5, while additivity is shown as FIC index of >0.5 ≤ 4 and antagonism is shown as an FIC index of >4. FIC index was an average of two independent experiments [36].

### 4.5. Time-Killing Assay

The in vitro bactericidal activities of imipenem and amikacin were evaluated by time-kill curves. The tested resistant strains used in this assay were *P. aeruginosa* isolate (wound no. 5) resistant for both drugs, *P. aeruginosa* isolate (wound no. 3) resistant for amikacin only, and *P. aeruginosa* isolate (wound no. 2) resistant for imipenem *only*. The test was performed five times using concentrations of 0.5 × MIC, 1 × MIC, 2 × MIC, and 4 × MIC, in both single-drug and combination studies. Equal volume of tested concentration was mixed with 1 MacFarland of bacterial suspension (final bacterial concentration 0.5 MacFarland) incubated to 24 h at 37 ℃ at time interval (0, 2, 4, 8, 12, and 24 h) from incubation by plating 10-fold dilutions on sheep blood agar [37]. Bacteriostatic activities were defined as the presence of ≥2 log_10_, but <3 log_10_ reductions, and bactericidal activities as the presence of ≥3 log_10_ reductions in CFU/mL at 24 h, relative to the initial inoculum, while synergy was defined as a 2 log_10_ decrease in CFU/mL when using the drug combination, relative to the most active component alone. All experiments were performed five times [38].

### 4.6. Gene Expression of bla _IMP_ and aac(6′)-Ib Combination by Real-Time PCR

*Bla-_IMP_* and *aac(6′)-Ib* are the most prevalent resistant genes for imipenem and amikacin in *P. aeruginosa* strains. Quantitative real-time PCR (q RT-PCR) was used for the detection of gene expression of *bla _IMP_* and *aac*(*6′*)*-Ib* with 16srDNA house-keeping gene as a control previously mentioned in Table 1. The isolates used for gene expression were: *P. aeruginosa* isolate (wound no. 5) resistant for both drugs harboring *aac*(*6′*)*-Ib* and *bla _IMP_, P. aeruginosa* isolate (wound no. 3) resistant for amikacin harboring *aac(6′)-Ib only,* and *P. aeruginosa* isolate (wound no. 2) resistant for imipenem harboring *bla _IMP_* only.

The measurement of gene expression of the two genes in the resistant isolate for both drugs *P. aeruginosa* isolate (wound no. 5) was done before and after treatment with the antibiotic. The concentrations used in the treatment were in the dose under the MIC value equal to 0.25 × MIC and 0.5 × MIC of both drugs to allow the bacterial growth with induction of antibiotic resistance [39]. Extraction of bacterial RNA was done according to RNeasy Mini Kit instructions using Quantitect SYBR green PCR kit (Qiagen, Germany). The RT-PCR was performed in 25 μL reaction mixture consisting of 2× QuantiTect SYBR Green PCR Master Mix (12.5 μL), reverse transcriptase (0.25 μL), 0.5μL of each forward (20 pmol) and reverse primers (20 pmol), RNase free water (8.25 μL), and template RNA (3 μL).

The cycling conditions were the same as indicated in conventional PCR table (7). Amplification curves and CT values were determined by the strata gene M × 3005 P software. To estimate the variation of gene expression on the RNA of the different samples, the CT of each sample was compared with that of the control group according to the “ΔΔCt” method stated by Yuan, et al. [40]. Dissociation curves were compared between different samples to exclude false-positive results.
Whereas ΔΔCt = ΔCt _reference_ – ΔCt _target_
ΔCt _target_ = Ct _control_ – Ct _treatment_ and ΔCt _reference_ = Ct _control_ − Ct _treatment_

### 4.7. Scanning Electron Microscopy (SEM)

Tested bacteria cells of *P. aeruginosa* isolate (wound no. 5) resistant to both drugs were suspended in a saline solution containing 0.2%Tween-80 and with amikacin alone, imipenem alone, and in combination at 2× MIC incubated at 37 °C. After 24 h, the bacterial cells were centrifuged at 8000 rpm for 15 min. The bacterial cells were then washed with 0.1 M tris-acetate buffer (pH 7.1), fixed in tris-acetate buffer containing 1.5% glutaraldehyde, and then freeze-dried. Each bacterial culture was observed by SEM (Hitachi, Japan) at magnifications of 10,000, 7500, and 15,000×. The bacterial cell suspension in saline with no antibiotics treatment served as a negative control [41].

### 4.8. In Vivo Studies 

Forty-week-old male CD-1 mice (in 4 groups, each group contain 5 mice), weighing 30 to 35 g were used. Treatment with imipenem, amikacin, and imipenem+amikacin combination started 3 h after intra-peritoneal infection, with 0.5 McFarland (1.5 × 10^8^ CFU/mL) of *P. aeruginosa* isolate (wound no. 5) resistant for both drugs, and the treatment lasted for 24 h. [36]. The animals were maintained in accordance with the recommendations of the Guidelines for the Care and Use of Laboratory Animals, and the experiments were approved by the Animal Care. Mice were treated by intraperitoneal injections with amikacin 15 mg/kg every 8 h, imipenem 40 mg/kg every 4 h, and amikacin and imipenem in combination (doses and intervals were the same as in monotherapy). These doses were chosen to simulate the human serum levels achieved with the commonly used doses. The negative control group received sterile saline intraperitoneal. The positive control and test groups received intraperitoneal bacteria; bacteremia was confirmed by isolating causative organisms on selective cetrimide agar plates. Blood samples (20–50 µL) were taken from the tail vein of selected mice in each group at 3, 11, and 27 h after infection. On Mueller–Hinton agar plates, 10 µL aliquots of 7 serial dilutions were plated and were incubated overnight at 37 °C for CFU determination [36].

### 4.9. Statistical Analysis

The collected data were coded, tabulated, and statistically analyzed using SPSS program (Statistical Package for Social Sciences) software version 25. Graphical presentation was done by using Microsoft office Excel 365 software. Descriptive statistics were done for parametric (normally distributed) quantitative data by mean, standard deviation (SD). Distribution of the data was done by Shapiro Wilk test analyses were done for parametric quantitative data between different groups or different concentrations using one-way ANOVA test followed by post hoc Tukey’s analysis between each two groups or each two concentrations. Analyses were done for parametric quantitative data between different times using repeated measures ANOVA test followed by post hoc LSD analysis between each two times. The level of significance was taken at (*p* value < 0.05).

## 5. Conclusions

In conclusion, the combination of imipenem plus amikacin suppressed the resistance against carbapenem and/or aminoglycoside in *P. aeruginosa* strains. So, their combination viewed synergistic bacterial killing and potential antimicrobial activity. The in vivo study agreed with the results obtained in vitro. Imipenem and amikacin combination could be an option in the treatment of MDR or XDR *P. aeruginosa*. Thus, future evaluation of this combination in dynamic infection models is ensured to provide guidance in treatment against infections caused by complicated treated *P. aeruginosa* infections. The reason for choosing these drugs was the lack of clinical studies which evaluate the efficacy of amikacin and imipenem combination, and only very limited cases published about the effect of aminoglycoside plus carbapenem combination therapy.

## Figures and Tables

**Figure 1 antibiotics-10-01429-f001:**
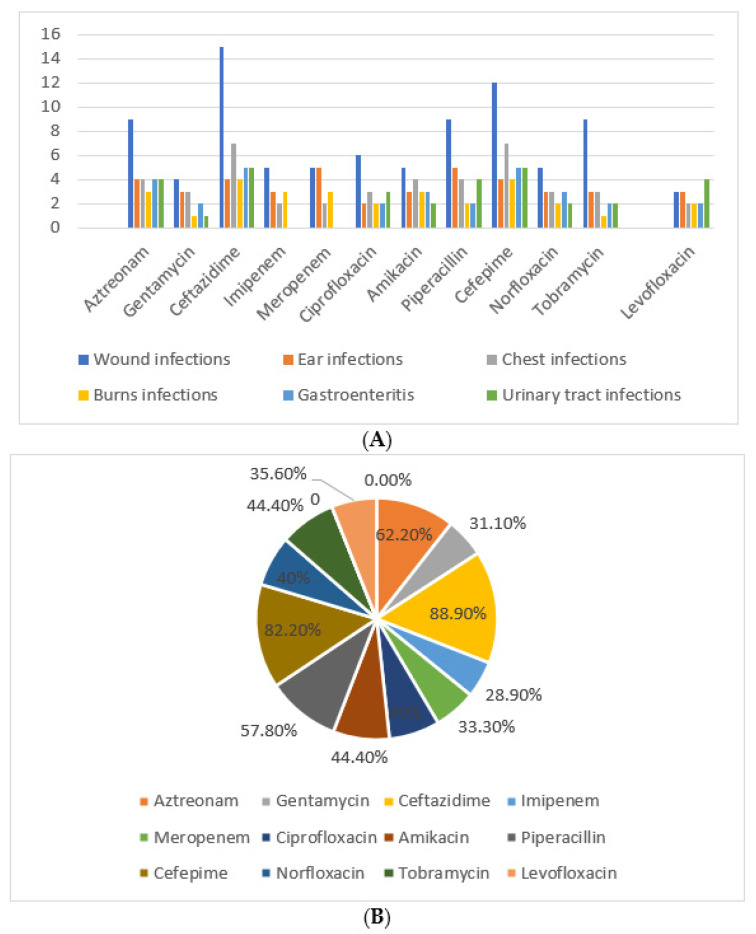
(**A**): Resistance pattern of *P. aeruginosa* isolates among the different infections. (**B**): Percent of antibiotic resistance correlated to 45 *P. aeruginosa* isolates.

**Figure 2 antibiotics-10-01429-f002:**
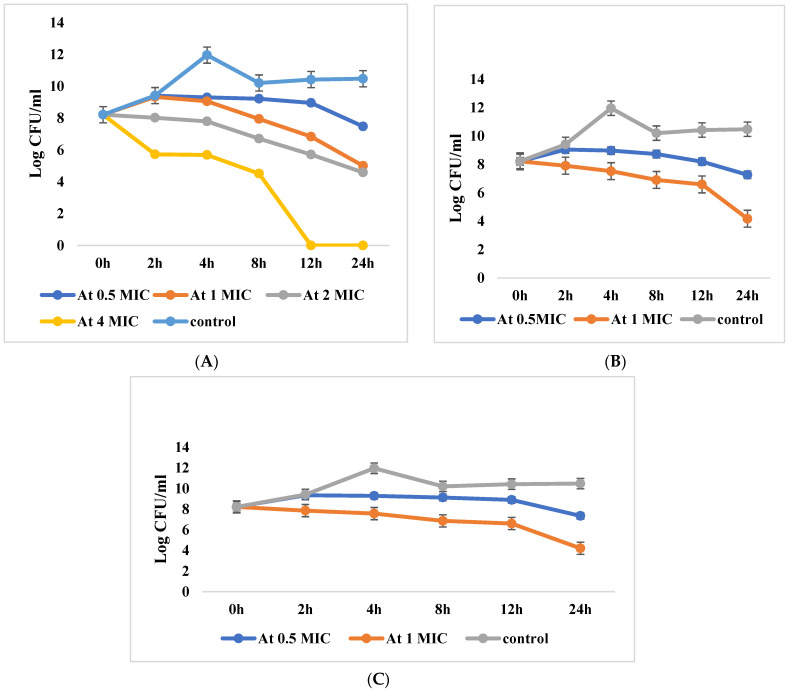
(**A**) Killing curves of imipenem and amikacin resistant *P. aeruginosa* incubated without antibiotic (control), with mikacin + imipenem at different concentrations (0.5 × MIC, 1 × MIC, 2 × MIC, 4 × MIC). (**B**) Killing curves of imipenem-resistant *Pseudomonas aeruginosa* incubated without antibiotic (control), with amikacin + imipenem at different concentrations (0.5 × MIC, 1 × MIC). (**C**) Killing curves of amikacin resistant *P. aeruginosa* incubated without antibiotic (control), with amikacin + imipenem at different concentrations (0.5 × MIC, 1 × MIC).

**Figure 3 antibiotics-10-01429-f003:**
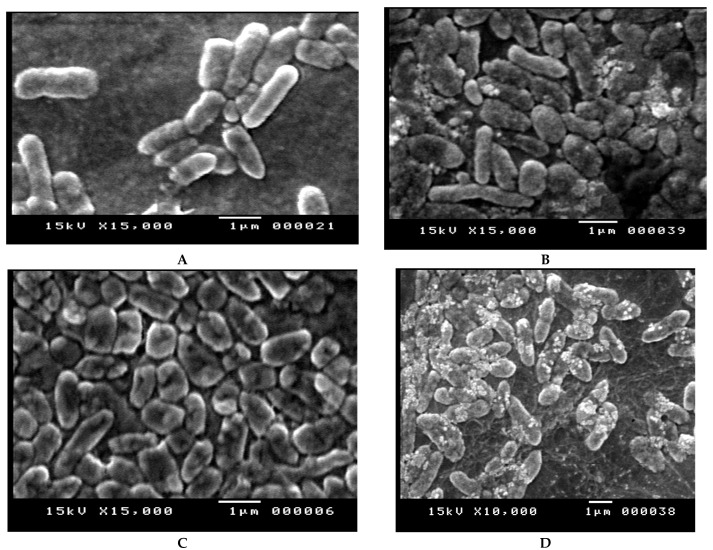
(**A**) SEM image of *P. aeruginosa* control; (**B**) SEM image of *P. aeruginosa* treated with imipenem; (**C**) SEM image of *P. aeruginosa* treated with amikacin; and (**D**) SEM image of P. aeruginosa treated with amikacin/imipenem.

**Figure 4 antibiotics-10-01429-f004:**
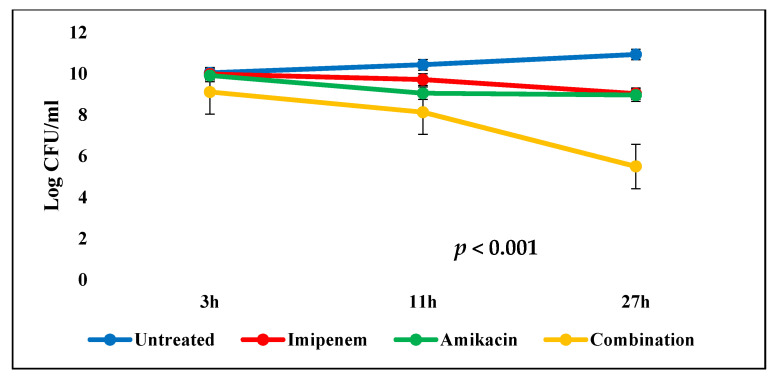
Mean blood bacterial counts (error bars represent standard deviations) observed in untreated mice and in those receiving amikacin (15 mg/kg every 8 h), imipenem (40 mg/kg every 4 h), and amikacin (15 mg/kg every 8 h) plus imipenem (40 mg/kg every 4 h). *p* value < 0.001 indicates significant difference between the four groups.

**Table 1 antibiotics-10-01429-t001:** Prevalence of isolated bacteria among different infections.

Source of Infections	Total Number of Isolates	*E. coli*	*P. aeruginosa*	*Proteus* spp.	*Klebsiella* spp.	*A. baumannii*
Wounds	78	29	17	24	5	3
Ear infections	10	3	6	1	-	-
Burns	15	5	5	5	-	-
Chest infections	22	7	8	-	5	2
Urinary tract infections	13	8	5	-	-	-
Gastroenteritis	12	8	4	-	-	-
Total (%) *	150 (100)	60 (40)	45 (30)	30 (20)	10 (6.67)	5 (3.33)

* percent were corelated to total number of isolates.

**Table 2 antibiotics-10-01429-t002:** Distribution of MIC, MIC_90_, and MIC_50_ of amikacin among the isolated *P. aeruginosa*.

No.	MIC(µg/mL)													MIC_90_	MIC_50_	R	% *
	0.25	0.5	1	2	4	8	16	32	64	128	256	512	1024	256	8		
45	0	0	6	6	6	7	4	3	3	3	3	2	2			13	28.9

* Percent was correlated to the number of *P. aeruginosa*.

**Table 3 antibiotics-10-01429-t003:** Distribution of MIC, MIC_90_, and MIC_50_ of imipenem among the isolated *P. aeruginosa*.

No.	MIC(µg/mL)													MIC_90_	MIC_50_	R	% *
	0.25	0.5	1	2	4	8	16	32	64	128	256	512	1024	256	2		
45	0	0	20	12	0	0	0	0	5	2	2	2	2			13	28.9

* Percentage was correlated to the number of *P. aeruginosa.*

**Table 4 antibiotics-10-01429-t004:** The combined effect between imipenem and amikacin against selected resistant *P. aeruginosa* isolates.

*P. aeruginosa*	MIC (µg/mL)				FIC_index_	Outcome
	Amikacin	Imipenem	Amikacin + Imipenem			
Wound isolate no. 5	1024	1024	32	32	0.33	Synergistic
Ear isolate no. 3	1024	256	32	32	0.4	Synergistic
Burn isolate no. 3	512	256	4	1	0.011	Synergistic
Chest infection isolate no. 1	1024	256	32	8	0.062	Synergistic

**Table 5 antibiotics-10-01429-t005:** Gene expression for resistant *P. aeruginosa* after treatment.

*P. aeruginosa* Isolate No.	*16S rDNA*	*bla _IMP_*		*aac(6′)-Ib*	
	CT	CT	Fold Change	CT	Fold Change
P Control	20.51	21.94	-	23.62	-
P1	21.70	20.10	8.1681	21.84	7.8354
P2	20.45	Nd		19.79	13.6422
P3	21.08	19.15	10.2674	Nd	
P4	21.13	20.37	4.5631	21.83	5.3147
P5	20.78	19.76	5.4642	21.90	3.9724

P1: *P. aeruginosa* isolate wound no. 5 resistant for both drugs. P2: *P. aeruginosa* isolate wound no 3 resistant for amikacin harboring *aac(6′)-Ib* only. P3: *P. aeruginosa* isolate wound no 2 resistant for imipenem harboring *bla _IMP_ only. P4:*
*P. aeruginosa* isolate wound no. 5 after treatment with 0.25 × MIC of amikacin + 0.5 × MIC of imipenem. P5: *P. aeruginosa* isolate wound no. 5 after treatment with 0.25 × MIC of imipenem + 0.5 × MIC of amikacin.

**Table 6 antibiotics-10-01429-t006:** Comparisons between different groups of *Pseudomonas aeruginosa* in different times.

Time	Untreated (I)	Imipenem (II)	Amikacin (III)	Combination (IV)	*p* Value
3 h	10.04 ± 0.03	9.98 ± 0.02	9.91 ± 0.03	9.11 ± 0.02	<0.001 *
11 h	10.43 ± 0.01	9.71 ± 0.05	9.05 ± 0.03	8.13 ± 0.03	<0.001 *
27 h	10.93 ± 0.04	9.03 ± 0.03	8.96 ± 0.03	5.50 ± 0.52	<0.001 *

* Significant *p*-value: <0.001.

**Table 7 antibiotics-10-01429-t007:** The sequences of the primers used in this study.

Gene	Primer Sequence (5’-3’)	Annealing Temperature (° C)	Product Size (bp)	Reference
*bla _IMP_*	F:CATGGTTTGGTGGTTCTTGT	59	488	[33]
	R:ATAATTTGGCGGACTTTGGC			
*aac(6′)-Ib*	F:AGTACTTGCCAAGCGTTTTAGCGC	51	365	[34]
	R:CATGTACACGGCTGGACCAT			
*16S rDNA*	F:GACGGGTGAGTAATGCCTA	55	618	[35]
	R: CACTGGTGTTCCTTCCTATA			

## Data Availability

Not applicable.

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
