# Peer review of "Effect of Imipenem and Amikacin Combination against Multi-Drug Resistant Pseudomonas aeruginosa"

_antibiotics, 2021, doi:10.3390/antibiotics10111429_

Round 1
Reviewer 1 Report
1. The overall research goal and objectives should have been elaborated and the hypothesis was not clear to me.
2. In the introduction, there should have a better explanation about why this study is scientifically important than the previously published articles.
3. In the introduction, the Rationale for using combination therapy with carbapenem and aminoglycoside should have been more extensive.
4. Line 51- 52 may need a reference.
4. Figure 1 seems not to correlate or be clear with the description of section 2.2. The Y-axis of figure 1 is undefined.
5. In Table 3, the definition of resistance to imipenem (MIC≥8) needs to be clarified. should have given a rationale to choose MIC≥8.
6. Figure 3, using more diluted bacterial culture could have given improved images
Author Response
Comment 1
- The overall research goal and objectives should have been elaborated and the hypothesis was not clear to me.
Reply 1
research goal and objectives have been rewritten in more details
Comment 2
2.In the introduction, there should have a better explanation about why this study is scientifically important than the previously published articles.
Reply 2
To the best of our knowledge, there are no data relating to whether a combination of imipenem with an aminoglycoside is better than imipenem alone against P. aeruginosa. So, this study is important due to the emergence of MDR, XDR and PDR P. aeruginosa which is considered a great challenge to be treated by many antibiotics including imipenem due the pathogen nature and their intrinsic resistance to many antibiotics. Also, their ability to acquire multiple imported resistance mechanisms on mobile genetic elements or mutations. (Added in the line 207)
Comment 3
3.In the introduction, the Rationale for using combination therapy with carbapenem and aminoglycoside should have been more extensive.
Reply 3
The rationale for using combination therapy with carbapenem and aminoglycoside have been added in more details (line 86 and second paragraph in discussion)
Comment 4
4.Line 51- 52 may need a reference.
Reply 4
reference in line 51-52 have been added
Comment 5
- 5. Figure 1 seems not to correlate or be clear with the description of section 2.2. The Y-axis of figure 1 is undefined.
Reply 5
Figure 1 have been changed and separated in two graphs
Comment 6
- In Table 3, the definition of resistance to imipenem (MIC≥8) needs to be clarified. should have given a rationale to choose MIC≥8.
Reply 6
definition of resistance to imipenem (MIC≥8) and their rationale use it was chosen according to CLSI 2018 and it was mentioned before in material and method
Comment 7
- 7. Figure 3, using more diluted bacterial culture could have given improved images
The bacterial culture was diluted but centrifugation lead to the aggregation of bacterial cells and their appearance condensed. Also, images appeared according to the resolution of the electron microscope

Reviewer 2 Report
Antibiotic resistance is a growing global menace. In this manuscript, Farhan et al. investigate the in-vitro and in-vivo efficacy of imipenem and amikacin combination against P. aeruginosa. This work will be of broad interest to readers of the antibiotic journal. However, some items need to be addressed before accepting.
- My primary concern is about Figure 1. The piperacillin/tazobactam and ofloxacin treatments were conducted for the antimicrobial susceptibility test (Line 244). However, they were not included in Figure 1. In addition, authors are suggested to remove the brown bars (% of resistance) from this figure and plot in a new figure. I could not find any brown bar reaching 28.9 % (Line 8).
- In the introduction section, it would be nice to add description of the functions of the imipenemase aminoglycoside 6'-N-acetyltransferase.
- Line 93-96: it is critical to know the percentage of P. aeruginosa, which carries both blaIMP and aac (6’-)-Ib gene.
- The experiment design of the gene expression assay is very confusing. I understand the treatment for P4 is 0.25*MIC of amikacin+0.5*MIC of imipenem, and the treatment for P5 is 0.25*MIC of imipenem+ 0.5* MIC of amikacin. However, what are the treatment for P, P1, P2, and P3?
- Line 127 and 129: change “MC” to “MIC.”
- Capitalize “mic” in Figure 2. a/b/c
- Consistent font size needs to be applied for Line 114,133, 134, 295,296 and 297.
Author Response
Comment 1
My primary concern is about Figure 1. The piperacillin/tazobactam and ofloxacin treatments were conducted for the antimicrobial susceptibility test (Line 244). However, they were not included in Figure 1. In addition, authors are suggested to remove the brown bars (% of resistance) from this figure and plot in a new figure. I could not find any brown bar reaching 28.9 % (Line 8).
Reply 1
Regarding the piperacillin/tazobactam and ofloxacin treatments were conducted for the antimicrobial susceptibility test (Line 244), they were written by mistake and they have been removed.
About the figure 1,as suggested the % of resistance were separated in another figure (figure 1B).
Comment 2
In the introduction section, it would be nice to add description of the functions of the imipenemase aminoglycoside 6'-N-acetyltransferase.
Reply 2:
Sections describing the functions of the functions of the imipenemase and aminoglycoside 6'-N-acetyltransferase were added to the introduction.
Comment 3
Line 93-96: it is critical to know the percentage of P. aeruginosa, which carries both blaIMP and aac (6’-)-Ib gene.
Reply 3
The percentage of P. aeruginosa, which carries both blaIMP and aac (6’-)-Ib gene was added
Comment 4
The experiment design of the gene expression assay is very confusing. I understand the treatment for P4 is 0.25*MIC of amikacin+0.5*MIC of imipenem, and the treatment for P5 is 0.25*MIC of imipenem+ 0.5* MIC of amikacin. However, what are the treatment for P, P1, P2, and P3?
Reply 4
P : is the control that was sensitive to both drugs and did not carry the both genes , so it had no treatment.
P1: the isolate was resistant to both drugs and carrying both genes. Did not receive any treatment
P2: the isolate was resistant for amikacin and harboring aac(6’)-Ib only. Did not receive any treatment
P3: the isolate resistant for imipenem harboring blaIMP only. Did not receive any treatment
P4 and P5 are the isolate no P1 after receiving the different treatment.
So that, the genes expression was detected in case of the presence of both genes or in the presence of only one gene without treatment to be compared with expression after treatment.
Comment 4
Line 127 and 129: change “MC” to “MIC.”
Reply 4
The error was resolved.
Comment 5
Capitalize “mic” in Figure 2. a/b/c
Reply 5
The mic” in Figure 2. a/b/c was capitalized as recommended
Comment 6
Consistent font size needs to be applied for Line 114,133, 134, 295,296 and 297.
Reply 6
The font size is fixed in the mentioned positions
